# Bridging Background Knowledge Gaps in Translation with Automatic Explicitation

**HyoJung Han**
Computer Science
University of Maryland
hjhan@cs.umd.edu

**Jordan Boyd-Graber**
CS, UMIACS, iSchool, LCS
University of Maryland
jbg@umiacs.umd.edu

**Marine Carpuat**
Computer Science, UMIACS
University of Maryland
marine@cs.umd.edu

## Abstract

Translations help people understand content written in another language. However, even correct literal translations do not fulfill that goal when people lack the necessary background to understand them. Professional translators incorporate *explicitations* to explain the missing context by considering cultural differences between source and target audiences. Despite its potential to help users, NLP research on explicitation is limited because of the dearth of adequate evaluation methods. This work introduces techniques for automatically generating explicitations, motivated by WIKIEXPL[1]: a dataset that we collect from Wikipedia and annotate with human translators. The resulting explicitations are useful as they help answer questions more accurately in a multilingual question answering framework.

## 1 The Best Translations Go Beyond Literal Meaning

A good translation is one that literally conveys the correct meaning of every word in one language in another language. . . and this is what is rewarded by translation metrics like BLEU. However, the best translations go beyond the literal text (Snell-Hornby, 1990, 2006), viewing translation as means to an end: enable a communicative act in the target language. This requires "going beyond" strict equivalency to consider underlying presuppositions of the respective source and target cultures (Nida and Taber, 1969, 2003).

As a result, professional translators use a technique called "explicitation", an explicit realization of implicit information in the source language because it is apparent from either the context or the situation (Vinay and Darbelnet, 1958, 1995). One of the main motivations of explicitation is to convey the necessary background knowledge

---

[1]We release our dataset at https://github.com/h-j-han/automatic_explicitation.

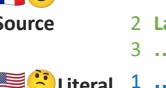
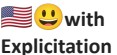

| | | |
|---|---|---|
| 🇫🇷🙂 Source | 1 | …frère de Dominique de Villepin… |
| | 2 | La veille de Noël 1800, sachant qu'… |
| | 3 | …l'attentat contre Charlie Hebdo… |
| 🇺🇸🙁 Literal Translation | 1 | …brother of Dominique de Villepin … |
| | 2 | The day before Christmas 1800, knowing that |
| | 3 | …the attack against Charlie Hebdo… |
| 🇺🇸🙂 with Explicitation | 1 | …brother of **the former French Prime Minister** Dominique de Villepin … |
| | 2 | On Christmas Eve in 1800, **amid the French Revolution**, |
| | 3 | …the attack against **the French satirical newspaper** Charlie Hebdo… |

Figure 1: Examples of French-English WIKIEXPL. Translators compensate for background knowledge gaps with explicitation in the target language. Here, while underlined parts do not need to be introduced in the French-speaking world, English speakers may need additional information, as in the red colored text, to enhance their understanding. Flags represent national identity and its associated cultural milieu rather than language.

that is generally shared among the source language speaking communities (Séguinot, 1988; Klaudy, 1993). Since the target language audience does not share the same background and cultural knowledge, translators often give an explanatory translation for better understanding, which Klaudy (1993, 1998) defines as "*pragmatic* explicitation". For example, the name "Dominique de Villepin" may be well known in French community while totally unknown to English speakers in which case the translator may detect this gap of background knowledge between two sides and translate it as "***the former French Prime Minister*** Dominique de Villepin" instead of just "Dominique de Villepin".

Despite its utility, research on explicitation is limited: 1) current automatic metrics of translation prefer translations that precisely convey the exact literal meaning of the original source (thus penalizing the addition of "extraneous information"); 2) the lack of labeled explicitation data hampers the study of automatic generation. Existing research including Hoek et al. (2015) and

Lapshinova-Koltunski and Hardmeier (2017) is confined to the explicitation of connectives or relational coreferences in discourse translation and lacks systematic strategies to automatically detect or generate the explicitation.

We take a new focus on explicitation and explore **whether making necessary implicit knowledge explicit can help downstream translation.** Thus, we generate explicitations for culturally relevant content, mimicking human translators. To capture when translators explicitate, we build a dataset (which we call WIKIEXPL) that collects the entities that are described differently across languages. This dataset allows us to identify entities that should be explained in translation and to generate those explicitations. Finally, we test whether our explicitations were useful through an automatic evaluation of the usefulness of explicitations with a multilingual question answering (QA) system, based on the assumption that good explicitations of culturally—for example—Polish entities will increase the accuracy of a QA system.

Although explicitation is very rare (0.3%) in the training corpus, the collected examples are adequate for developing an explicitation system that helps on a task that needs explicitation: question answering. Moreover, explicitation need not be onerous or expensive: a short phrase is enough to explain obscure entities.

## 2   What is Explicitation?

The term explicitation was first defined by Vinay and Darbelnet (1958) as "a procedure that consists in introducing in the target language details that remain implicit in the source language, but become clear through the relevant context or situation". Explicitation has been refined over the next decades: Nida (1964) use 'amplification' to refer to 'additions', which are derivable from the socio-cultural context to enhance readability or to avoid misunderstanding due to ambiguity. Blum-Kulka (1986) conduct the first systematic study of explicitation focusing on structural, stylistic, or rhetorical differences between the two languages, formulating an "*explicitation hypothesis*" which broadly states that a translation tends to be more explicit than a corresponding non-translation. Séguinot (1988), however, find that the definition of explicitation is limited and suggests that the term should be reserved for additions in a translated text that cannot be explained by those linguistic differences.

Klaudy (1993, 1996, 1998) elaborate on the idea and develop Blum-Kulka's work, proposing four types of explicitation: *obligatory*, *optional*, *pragmatic*, and *translation-inherent*. Subsequent studies (Baker, 1996; Øverås, 1998; Dimitrova, 2005) further refine these definitions and categories.

We focus on *pragmatic* explicitation in Klaudy's sense, the explicitation of implicit background information of the source side speaker where the main purpose is to clarify information that might not be available to the target audience. Other types of explicitation focus on synthetic or stylistic changes—for example, *obligatory* explicitation is mainly driven by difference of syntactic and semantic structure (e.g., different functions of prepositions and inflections) while *optional* is by fluency and naturalness (cf. translationese) (Klaudy, 1993, 1996). The main motivation of *pragmatic* explicitation, on the other hand, is to produce a well-suited translation aimed at a target audience to enable a communicative act (Snell-Hornby, 2006) by bridging the general knowledge gap.

We study automatic explicitation as one of the various efforts to accommodate not only linguistic but also cultural diversity, and further benefit the users of machine translation (MT) and cross-cultural NLP systems on a larger scale (Hershcovich et al., 2022; Dev et al., 2023).

## 3   Building the WIKIEXPL Dataset of Explicitations

This section describes how we collect and examine naturally-occurring explicitations in bitexts commonly used as MT training data. The resulting WIKIEXPL corpus lets us reason about *when* explicitation is necessary and lets our automatic explicitation method learn *how* to generate it.

### 3.1   How to Find Explicitation

Finding explicitation in the parallel text is a daunting task, so we first need to triage candidates of possible explicitation. To find explicitation examples, we follow the *explicitation hypothesis* (Blum-Kulka, 1986) and the traits of *pragmatic* explicitation (Klaudy, 1993, 1998) mentioned in Section 2 and assume the following properties of explicitation in our search[2]:

---

[2]In the Wikipedia multilingual data, we do not know whether a text and its aligned equivalents in other languages were generated by translation, from scratch in each language, or through some editing process that mixes the two. We set the direction of translation and find the explicitation under

1. Explicitations are part of unaligned token sequences: an unaligned segment in the target sentence could be an explicitation, as the *explicitation hypothesis* broadly states that a translation tends to be more explicit than a corresponding non-translation (Blum-Kulka, 1986; Pym, 2005).

2. Explicitations are close to named entities: the unaligned segment could be an explicitation if there is a named entity near the segment while its content is related to the entity and helpful for bridging a background knowledge gap between source and target language users. The gap is more likely to be significant if the entity is more specific to the background or culture of the source audience.

3. Explicitations are more likely for culturally distant entities: a major shift of some property values conditioned on one language to another could indicate the boundness of an entity to a specific language community. For example, the Wiki page for "Dominique de Villepin" is closer to pages of French-speaking than English-speaking countries in the relative relational distance.

Based on these assumptions, we develop a process to detect explicitation in bitext and decide whether the given entity needs explicitation or not.

As a main source of explicitation examples, we choose a not-too-clean parallel corpus from Wikipedia that naturally includes unaligned segments and divergence, as explicitation is by definition a non-literal translation making an implicit detail in the source explicit in the target, and thus introducing unaligned content in one of the languages. If the parallel corpus is too "clean" or too parallel, it is more likely to contain literal translations rather than explicitation examples.

Overall, building the WIKIEXPL dataset takes three main steps. First, we process the bitext and detect potential explicitation pairs of unaligned segments and entities (Sec 3.2). Secondly, we decide if the entity in the pair needs explicitation, resulting in the selection of candidates among the pairs (Sec 3.3). Lastly, we present extracted candidates

to a human translators for the final explicitation annotation (Sec 3.4).

## 3.2 Detecting the Explicitation in Bitext

We seek to find instances of explicitations candidates. We first find unaligned segments $u$ via word alignment, then we pair segment $u$ with the closest entity $e$ identified by named entity recognizer (NER). Next, we determine whether the segment $u$ is likely an explicitation of entity $e$ by checking if a pair of the segment $u$ and the entity $e$ are nearly positioned within the sentence and related. Formulation of detection is in Appendix, Algorithm 1.

## 3.3 Deciding If Explicitation is Needed

The ability to identify if the entity needs explicitation is the key to both detecting the explicitation example and automating the explicitation process. Since our focus is on culturally-specific explicitation, we need to algorithmically define what makes an entity specific to a socio-linguistic context.

Given a relational knowledge base (KB) graph, this can be implemented as the number of hops from an entity to source and target language-speaking countries. For instance, "Dominique de Villepin" is one hop away from France in Wikipedia, but multiple hops away from English-speaking countries.

A complementary method is to compare the number of incoming links to the Wikipedia page in a given language,[3] and the length of the Wikipedia page in a given language,[4] as these indicate the popularity of pages in a given language.[5]

We implement each of these properties and measure whether the property values prop of the given entity $e$ conditioned on each language $l$ of bitext pairs, and check if the shifts from the source language $l_{src}$ to target language $l_{tgt}$ are above the threshold $\tau$:

$$\mathsf{prop}(e \,|\, l_{src}) - \mathsf{prop}(e \,|\, l_{tgt}) > \tau \qquad (1)$$

An entity $e$ that passes all of these checks is considered to be strongly associated with the source language community. For example, if the property shift of the closeness (negative number of hops in KB) and normalized length in Wikipedia page is

---

the assumptions we made based on the *explicitation hypothesis*. What we focus on is how some entities are discussed differently in the language of their original culture vs. another language and using that information to design explicitation strategies. More details in Limitations.

[3] https://linkcount.toolforge.org/api
[4] https://{lang}.wikipedia.org/w/api.php
[5] We standardize (zero mean and unit variance) the value of both properties on each language within the entities in extracted WikiMatrix sentence pairs to account for the different offsets within a language.

meaningful enough, then the entity is considered as culturally bounded entity.

We additionally exclude entities that are well known globally from our annotation effort even if they are bound to a source community (e.g., the Eiffel Tower in Paris), as entities that are well-known globally are less likely to require explicitation. We use the number of languages in which a Wikipedia page is available for the entity to measure how well-known it is. Formulation of decision is in Appendix, Algorithm 2.

### 3.4 Annotation Framework for WIKIEXPL

To design explicitation models, we need ground truth examples and thus ask humans to manually verify the candidates automatically extracted from bitext as described above.

We present the candidates to the human annotators and let them label 1) whether the candidates are explicitation and 2) the span of explicitation. Annotators categorize unaligned words into "Additional Information", "Paraphrase (Equivalent and no additional info)", or "Translation Error/Noise" with a focus on the "Additional Information" class that potentially contains explicitation. Annotators then determine if explicitation is present by assessing if the additive description explicitly explains implicit general knowledge of the source language to the target language speakers. If confirmed, they mark the explicitation span in both source and target sentences and provide an optional note to justify their decision. Examples for each of categories and more details of the annotation framework are in Appendix B and Figure 8a.

Within the candidates, there are multiple nations within a single linguistic milieu, particularly in the case of French and Spanish. The annotators mark candidates that come from the same country as they do, since the precision and consensus of explicitation might be sub-optimal if, for instance, we assign French entities to Canadian annotators. All annotators are translators who are fluent in both English and the source language.

Each example is annotated by three annotators and we assign the label based on the majority vote. We consider the candidates as final explicitation if two or more annotators agree.

### 3.5 Experiment Settings

For the noisy parallel data, we use Wikimatrix (Schwenk et al., 2021) and extract the fr/pl/es-en pairs around the `threshold` of 1.051 to 1.050 for

| Source Language | French | Polish | Spanish |
|---|---|---|---|
| WikiMatrix | 29826 | 21392 | 28900 |
| Candidates | 791 | 245 | 307 |
| Top 1 country | 🇫🇷 | 🇵🇱 | 🇪🇸 |
| Annotated | 460 | 244 | 220 |
| Average $\kappa$ | 0.66 | 0.72 | 0.74 |
| At Least one vote | 236 | 111 | 73 |
| Explicitation | 116 | 67 | 44 |

Table 1: WIKIEXPL Construction Statistics. (*pragmatic*) Explicitations are rare in the noisy parallel corpus. France, Poland, and Spain are the country that is most frequently associated with the candidate entities for each source language. Candidates are shown to three annotators, and labeled as true explicitation by majority votes from the annotators. Cohen (1960)'s $\kappa$ coefficient shows high agreement among annotators.

French and Spanish, 1.052 to 1.050 for Polish.[6] We use WikiNEuRal (Tedeschi et al., 2021) for NER and mGENRE (De Cao et al., 2022) to get the Wikidata id for named entities. We ensemble the results of alignment tools, SimAlign (Jalili Sabet et al., 2020) and awesome-align (Dou and Neubig, 2021) to find un-aligned segments in bitext. For proximity, we decide that a segment is near an entity if it is within three words distance. We define the distance as a difference in the index of the tokenized words. For the relatedness between a segment and an entity, we check if a segment is in the content of an entity fetched from Wikipedia.

For the decision algorithm, we set the threshold as 1 for the property of the closeness (negative number of hops from entity to given language speaking country in KB), and implement it in practice by checking the existence of a direct relational link to the source country. We use this one property for extracting candidates, and after the data collection, we add two more properties of the noramlized number of incoming links and length of the Wikipedia page to optimize our decision function based on the collected data to have better accuracy. For entities that are well known globally, we exclude entities with Wikipedia pages in more than 250 languages.[7]

---

[6] https://github.com/facebookresearch/LASER/tree/main/tasks/WikiMatrix

[7] The hyperparameters are set on a development set drawn from a preliminary annotation stage, which uses the same framework in the main annotation but with non-expert annotators (volunteer graduate and undergraduate students). The method is applied to each language and it results in the same hyperparameter values.

| Type | Source | Target |
|---|---|---|
| Hypernym ($h$) | la Sambre | the Sambre ***river*** |
| Occupation/Title ($o$) | Javier Gurruchaga | ***showman*** Javier Gurruchaga |
| Acronym Expansion ($a$) | PP | ***People 's Party*** (PP) |
| Full names ($f$) | Cervantes | ***Miguel*** de Cervantes |
| Nationality ($n$) | Felipe II | Philip II ***of Spain*** |
| Integrated ($i$) | Dominique de Villepin | ***former French Prime Minister*** Dominique... |

Table 2: Examples of explicitation are categorized into five types. Integrated are examples with two or more types of explicitation. The boldface indicates added text by explicitation. The spans of added text are marked by annotators. More examples and full-sentence version available in Table 8 and Figure 8b.

### 3.6 Quantitavite Analysis on WIKIEXPL

Explicitation is quite rare (Table 1) which agrees with *pragmatic* explicitation statistics in Becher (2010). About 1–3% of sentences are explicitation candidates. France, Poland, and Spain are the country that is most frequently associated with the candidate entities for each source language. The overall ratio of the final explicitation example from the initial corpus is 0.2–0.4%.

The agreement among annotators is reliable ($\kappa \approx 0.7$) across the languages but suggests there is some subjectivity. About one-third to half of the candidates are marked as explicitation by at least one annotator.

### 3.7 Qualitative Analysis on WIKIEXPL

Among the collected data from all three languages, we analyze examples and categorize them into five types (Table 2): Hypernym, Occupation/Title, Acronym Expansion, Full names, and Nationality. A final type, Integrated, is for the examples with two or more types of explicitation. All the patterns we discovered are consistent with explicitation literature (Klaudy, 1996; Baumgarten et al., 2008; Gumul et al., 2017). The most common type in our collection is adding nationality information to the entity, especially for locations.

Realization of explicitation is diverse. The additional information could be accompanied by adjectives, prepositions, integrated into an appositive with commas, or in a parenthetical expression. Detailed descriptions on each type are in Appendix C. Additional examples of each type are available in Table 8, and the full-sentence version with the comments from the annotators are in Figure 8b.

## 4 Automating Explicitation

Previously, the only source of explicitation has been human translators. This section builds on the data from Section 3 to explore generating explicitations automatically.

### 4.1 Deciding if Explicitation is Needed

Abualadas (2015) contends translators judge the assumed target reader to decide if an explicitation is needed. Simulating such a decision process is challenging as it requires both instantiating a hypothetical reader and predicting what they know. In contrast, Section 3.3 simplifies this by providing explicitations for entities that are tightly bound to the source socio-linguistic context. We further optimize our decision function to have better accuracy by diversifying the types of properties and tuning the thresholds based on WIKIEXPL.(Section 3.5)

### 4.2 Generating the Explanation

We explore several forms in generating the explicitation: 1) SHORT: inserting one or two words after or before the entity, 2) MID: several words or a phrase integrated into the original translation in the form of appositives or parenthetical clauses 3) LONG: 1–3 sentences apart from the original translation text as a form of a footnote (examples in Table 3). Although SHORT and MID are in the examples of explicitation in Table 2 and LONG is not, such a long explanation is also considered explicitation and its usual surface manifestation would be a footnote (Gumul et al., 2017). We explore the validity of these three generation types of explicitation and seek to find the most effective one.

Our generation is grounded in Wikidata and Wikipedia—rather than free-form text generation—to prevent hallucinations and to control length or the type of explanation. For SHORT explicitations, we fetch a word from `instance of` or `country of` from Wikidata (cf. Hypernym, Title, and Nationality in Table 2). For MID, we fetch a description of the entity from Wikidata (mostly the Integrated in

| Type | Length | Form | Example of "Sambre," |
|---|---|---|---|
| SHORT | 1–2 words | Appositive | Sambre river, |
| MID | 3 words–a phrase | + Parenthetical | Sambre, river in France and Belgium, |
| LONG | 1–3 sentences | Footnote | Sambre (a river in northern France and in Wallonia, Belgium. It is a left-bank tributary of the Meuse, which it joins in the Wallonian capital Namur.) |

Table 3: Three types of generation by the length. The form of explicitation for SHORT and MID is appositive or parenthetical which directly integrates the additive description into the text, while LONG explanation is in the form of a footnote. Generation examples from our experiment in extrinsic evaluation are available in Figure 6.

| | Useful in target QA | Not Useful in target QA |
|---|---|---|
| Useful in source QA | Subject we don't know commonly | Subject is not popular in source community |
| Not Useful in source QA | Subject is well-known only in source community → **Need Explicitation!** | Subject is well-known globally |

Figure 2: How we check if our explicitations work: given source Polish questions, well-generated explicitations in English will improve English QA.

Table 2). For LONG type, we fetch three sentences from the first paragraph of Wikipedia.

## 5 Evaluating Explicitation

The evaluation of explicitation is challenging as how "useful" it is depends is subjective, it depends on what the hearer knows. We suggest two evaluations: intrinsic and extrinsic. —as each one has its own limitations but are complementary.

### 5.1 Intrinsic Evaluation

We ask the same human annotators who identified naturally occurring explicitations (Section 3.4) if our automatic explicitation is as useful as natural ones. We evaluate Polish to English translation, with users rating the English translation. The annotator rates both aspects of explicitation—whether the entity needed explicitation and the quality of the explicitation—with a three-step Likert scale: high, mid, and low. These are anchored with "not necessary" or "wrong explanation" at the low end and "appropriate and well-generate" or "necessary" at the high end. Additionally, we ask the annotator if the explicitation matches the surrounding context well: high (smoothly integrated), mid, and low (introduces a grammatical error).

### 5.2 Extrinsic Evaluation with Multilingual QA

Inspired by the studies using QA as extrinsic evaluation in cross-lingual settings (Tomita et al., 1993;

Krubiński et al., 2021), we use a multilingual QA framework to evaluate our automated explicitation.

A good explicitation adds necessary information that the original sentence lacks. If that new information helps a QA system get to the right answer, it suggests that the information was useful and well-targeted. We assume that the explicitation may not be helpful if the QA system is already knowledgeable of what is added by explicitation, and thus has minimal performance changes after feeding the questions with explicitation.

Under this assumption, we hypothesize that a well-generated explicitation will increase the accuracy of the target language QA task while being relatively less effective in the source language (Figure 2). For example, explicitation of a French cultural concept will be more useful to answer questions in a culturally distant language such as English than in the original French. In this QA setting, "usefulness" is clearer than in the intrinsic setting: usefulness is how much it improves QA accuracy, which could complement subjectivity in the intrinsic evaluation (Feng and Boyd-Graber, 2019).

To see the difference in the effectiveness between two QA tasks in source and target languages, we use parallel QA text. First, we identify the entity mention in both question texts and then see if it should be the target of explicitation with our decision function. If it is, we generate similar explanations for both languages and integrate them into each question text. Finally, we measure the effectiveness of explicitation in both languages and see the difference between the languages.

Specifically, we use Quizbowl setup (Rodriguez et al., 2019), which deals with incremental inputs and sequential decision-making, allowing us to examine whether the QA system is getting it right word by word. Compared to entire question accuracy in a standard QA setup, this approach allows us to analyze the effects of explicitation more pre-

|        | Num of Qs | Total Es | Explicitation |
|--------|-----------|----------|---------------|
| XQB-*pl* | 420     | 1009     | 62            |
| XQB-*es* | 144     | 598      | 116           |

Table 4: Statistics of XQB dataset for the evaluation. For the case of XQB-*es*, the 598 named entities are detected in both sides of Spanish and English pairs of 144 questions. Among 598 entities, our decision algorithm decides 116 entities need explicitation. The extraction rate of explicitation in the domain of QA dataset is higher than in the general domain.

| Decision | Type  | Generation | Integration |
|----------|-------|------------|-------------|
|          | SHORT | 0.63       | 0.79        |
| 0.71     | MID   | 0.82       | 0.92        |
|          | LONG  | 0.95       | _           |

Table 5: Intrinsic evaluation. 70% of automated explicitation are marked as valid decisions by human evaluators. The quality of generated content increases as the amount of added information gets larger. Integration of LONG is not considered as it is integrated into the form of a footnote.

cisely as its use case in evaluation of simultaneous interpretation (Han et al., 2022) and as in the main results (Section 7).[8]

### 5.3 Experiment Settings

**Dataset.** For the parallel QA dataset, we use the Cross-lingual Quizbowl test set (Han et al., 2022, XQB). XQB-*es* has 148 parallel Spanish to English question pairs and XQB-*pl* has 512 for Polish to English. One question usually consists of 3–5 sentences (Table 4). We use question pairs that have named entities recognized in both source and target languages text.

**Model.** Our multilingual QA system is LLAMA (Touvron et al., 2023) (7B for XQB-*es* and 13B for XQB-*pl*). This model is comparable or better without finetuning compared to RNN models trained on Quizbowl (Rodriguez et al., 2019) datasets (Appendix A Table 6). The input prompt is one full question, one partial question, answers, and simulated scores $\in [0, 1]$ as a guiding example, and append a real question at the end. An example of an input prompt is in Appendix, Figure 5. We set the output of the LLAMA model to have one guess and its confidence score, and use a threshold buzzer to decide buzz. The confidence threshold for the buzzer in EW is set to 0.4 for XQB-*pl* and 0.8 for XQB-*es*, which is fit to the original text without explicitation. We accept any of the synonyms from Wikidata in either the source or target language as a correct answer. We

set 30–50 character splits for the step size, usually having 30–32 splits for one question.

**Metric.** We adopt the same metrics, Expected Wins (EW) and EW with an oracle buzzer (EWO) as Rodriguez et al. (2019) and Han et al. (2022). Both metrics map the position of where a system answers a question to a number between 0 and 1 (higher is better). The difference is when the system provides an answer: EWO assumes a QA system can answer as soon as it predicts correctly as the top-ranked hypothesis while EW is a more conservative measure that requires not just producing an answer but also deciding whether to offer its guess as the answer (i.e., confidence estimation).[9] In both cases, the number represents the probability that the system will answer before a "typical" human from the Boyd-Graber et al. (2012) dataset, weighting early answers higher than later answers. Full Input Accuracy is a measurement with a whole text input of a question unlike EW or EWO.

## 6 Intrinsic Evaluation Results

Annotators evaluate explicitations in English questions of XQB-*pl* on decision, generation, and integration (setup in Section 5.1, example in Figure 6). To turn the Likert scale into a single number, we interpret high as 1, mid as 0.5, and low as 0. For the quality of generation and integration, we show the result for each generation type. Annotators score the explicitation decision 0.71 (Table 5). Negative examples (boldface as additional explanation added by explicitation) include too obvious ones (e.g., "Warsaw, ***the capital of Poland***") or those evident given the context of the sentence ("authors include the ***novelist*** Sienkiewicz").

The quality of generation is assessed highest on LONG type where the annotator evaluates about

---

[8]For consistent segmentation between the original question text and explicitation question text, we force the split at the boundary of the entity. Additional explanation by the explicitation is treated as the same single step with the entity, regardless of their additional length For example, "Sambre river" in explicitation will be the same step as "Sambre" in the original even though there is additional text. This may not be realistic for the acoustic setting where there is a time gap to deliver the added information, but we assume the text display setting where the additional text can be presented immediately.

[9]Further details of metrics are available in Rodriguez et al. (2019).

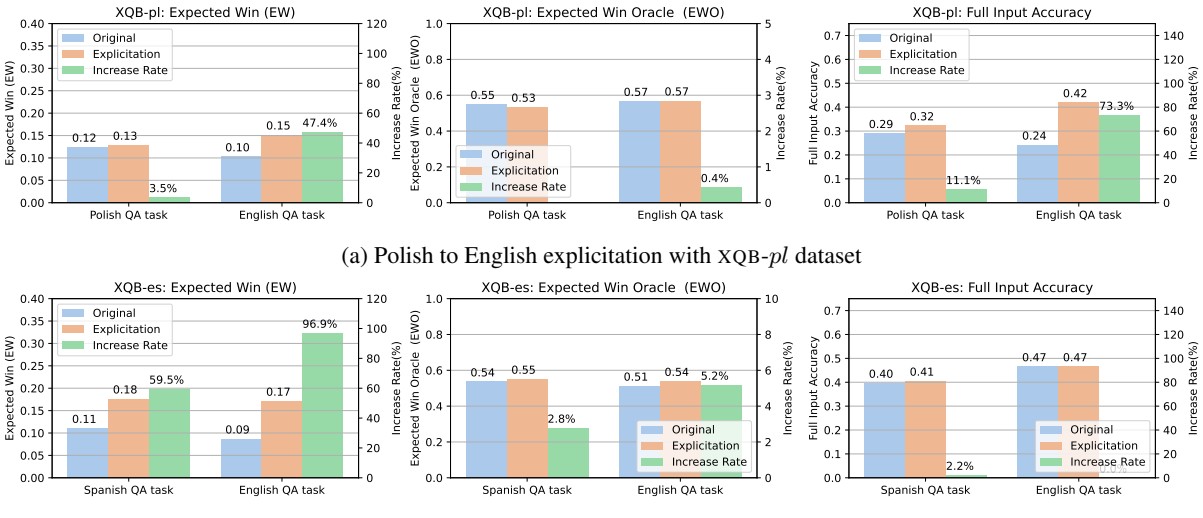

(a) Polish to English explicitation with XQB-*pl* dataset

(b) Spanish to English explicitation with XQB-*es* dataset

Figure 3: Extrinsic evaluation of automatic explicitation by comparison of the effect of explicitation. The original is the performance without explicitation. A higher increase rate in the English QA task indicates the effectiveness of our automatic explicitation methods. The generation type for all plots is MID.

95% of the generated explanations are appropriate and useful. The quality of generation decreases with shorter explanations. The added explanation should be easy for the target reader, but sometimes the explanation itself needs an explanation: for "Sejny, *city and urban gmina of Poland*", the annotators point out that "town" is more familiar than "gmina" to an English audience.

## 7 Extrinsic Evaluation Results

**Main Results.** Explicitation is more effective in English QA tasks than Polish on all metrics in XQB-*pl* (Figure 3a), which indicates that our decision algorithm effectively selects entities that need explicitation and that the added information is provides useful information (complementing the intrinsic evaluation in Section 6). XQB-*es* shows similar trends on EW and EWO while full input accuracy shows almost the same results after the explicitation in Figure 3b.

We attribute the different trends on full input accuracy for XQB-*pl* and XQB-*es* to different levels of difficulty of full-text questions: the questions that includes culturally bounded entity in XQB-*es* are easier than in XQB-*pl*. If a question is already easy, then additional information would not be very helpful and thus shows a small or no increase rate like in XQB-*es*. This is corroborated by the huge accuracy gap of 0.2 between XQB-*es* and XQB-*pl* in English QA. On the other hand, both in XQB-*pl* and XQB-*es* show great improvements of increase rate

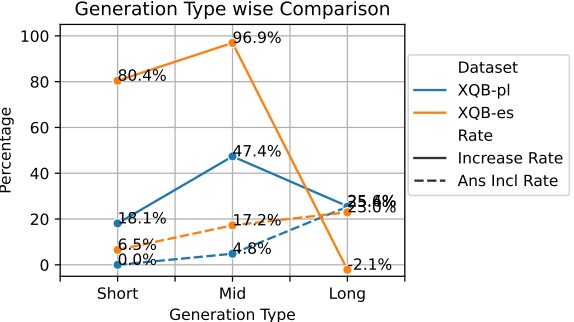

Figure 4: Increase rate of EW on English QA task by different types of explicitation generation, and the rate of answer inclusion. MID turns out to be most effective generation type for both XQB-*pl* and XQB-*es*. Answer inclusion rate increases as the length of additional explanation increases, while does not affect the increased rate of performance.

from English QA task to Polish/Spanish QA task on EW. Due to the structure of pyramidal difficulty within a single question and incremental input of Quizbowl setting, EW metric is able to capture the benefits of automatic explicitation in more sensitive way than the full input settings. Generally, answering the question early and correctly is harder than answering a question correctly given the full input, and explicitation in the middle of the question text could help QA system answer more quickly.

An additional comparison between explicitation and non-explicitation strengthens the validity of our decision algorithm in Appendix E, Figure 7.

**Generation Type Comparison and Answer Inclusion Rate.** We compare the generation type described in Section 4.2 and analyze the effect of answer inclusion in the explicitation (Figure 4). MID is the most effective type of explicitation, and longer explicitations are not necessarily more effective but are more likely to include the answer, which differs from what we observe in intrinsic evaluation where the longer explanation tends to have a better quality of explanation. The output example of each generation type within a XQB-*pl* question pair is available in Figure 6. (Further analysis in Appendix D, Table 7)

## 8 Related Work

**Explicitation in Contemporary Works.** Many of contemporary works with the term "explicitation" focus on discourse MT (Hoek et al., 2015; Webber et al., 2015) and mainly developed by Lapshinova-Koltunski et al. (2019, 2020, 2021). However, despite the broad coverage of the term, the explicitation in these studies are limited to insertion of connectives or annotations of coreference in the target side. Although the high-level concept of explicitation is common and our detection of explicitation starts similarly to Lapshinova-Koltunski and Hardmeier (2017) by finding alignment discrepancies, our focus is on different aspects of explicitation where the main purpose is to fill the gap of background knowledge between source and target sides. Krüger (2020) attempts to identify instances of explicitation in machine translated documents, while it deals with more general definition of explicitation rather than culturally-specific explicitation.

Explicitations can also be viewed as a form of divergence in meaning between source and target text. However prior work on detecting these based on cross-lingual semantic representations (Vyas et al., 2018; Briakou and Carpuat, 2020) target a much broader category of divergences than those that are culturally relevant.

Our work also relates to contemporaneous work on culturally aware MT. Yao et al. (2023) introduce a data curation pipeline to construct a culturally specific parallel corpus, and explore LLM prompting strategies to incorporate culturally specific knowlege into MT. Lou and Niehues (2023) introduce a semi-automatic technique to extract explanations based on Wikikpedia and alignment tools, similar to ours. Our work complements these studies by grounding the definition of explicitation in the translation studies literature, and by evaluating explicitations with both an intrinsic human evaluation and an extrinsic evaluation of their usefulness in multilingual QA.

**Elaboration and Clarification.** The explicitation of implicit background knowledge resembles text simplification and question rewriting, in the sense that these techniques make the text more accessible to targets, thus enhancing communications. Srikanth and Li (2021) and Wu et al. (2023) present elaborative text simplification where it adds the contents to elaborate the difficult concepts. Rao and Daumé III (2018, 2019) develop methods for generation and reranking clarification questions that ask for information that is *missing* from a given context. Elgohary et al. (2019) introduces the task and the dataset of question-in-context rewriting that rewrite the context-dependent question into a standalone question by making the context explicit. Ishiwatari et al. (2019) performs a task of describing unfamiliar words or phrases by taking important clues from both "local" and "global" context, where we have in common in the methods of generating description from Wikidata.

## 9 Conclusion and Future Work

We introduce techniques for automatic explicitation to bridge the gap of background knowledge between the source speaker and the target audience. We present an explicitation dataset, WIKIEXPL, extracted from WikiMatrix and annotated by human translators. We verify the effectiveness of our automatic explication with a direct evaluation by human translators and with extrinsic evaluation by using a multilingual QA and comparing its influence on different language tasks. Our automatic explicitation system is effective based on both intrinsic and extrinsic evaluation.

Future works include more closely simulating an explicitation decision and generation process of a human translator as it requires both instantiating a hypothetical reader and predicting what they might know and how to describe it while trying not to overdo it. Rather than machine QA systems, having human participants complete the QA task with and without explicitations will let us measure their usefulness more directly. Another extension would be adapting a speech-to-speech simultaneous interpretation format where the lengthy explicitation will be penalized, and thus taking the context into account to have less redundant output.

## Limitations

The number of explicitations samples we collected in WIKIEXPL is small, particularly when compared to the wealth of massively multilingual benchmarks that annotate relatively common language phenomena. Still, we argue that it provides a sufficient basis for a first study of cross-lingual explicitations, a relatively infrequent and understudied phenomenon, as a valuable resource for research on translation. Furthermore, we contend that the methodology we introduced can be used to expand it further.

We simplify the concept of culture by choosing one majority country to which the entities of candidate explicitation belong among extracted candidates from Wikidata (Section 3.5 and 3.6). For example, we choose France among many French-speaking countries as the number of entities from France is the largest among the extracted candidates. Then, we collect an explicitation example with French annotators as the accuracy and agreement of explicitation could be sub-optimal if we ask, for example, French entity to Canadian annotators. As a consequence, there is a possibile mismatch between languages and cultures that would degrade automated explicitation as the hyperparameters like the threshold of properties in Section 3.2 may not be optimal for different cultures.

The experiments are limited to one direction, into English. Our methods of automating explicitation and its evaluation focus on English speakers. We use simplified methods of integrating explicitation into a translation that mainly works for the English language, which could be further improved by improving fluency.

In finding the explicitation in Wikipedia, we set the direction of translation and collect the example base on the trait of explicitation that includes the unaligned tokens as discussed in Section 3.1. However, there is no information about the exact translation direction, the methods of translation, or even if it is a non-translation and just aligned equivalents that are generated from scratch independently in each language. For example, in the given bitext of French and English, we do not know if this is translated from French to English or English to French. Our focus is on how the same entity is presented differently in its original culture and in different language-speaking cultures, and this can be conducted without knowing the exact generation process of bitext. Better quality of explicitation could be collected via extracting the bitext that has a clear translation process.

Our decision and generation algorithm are restricted to named entity based on the assmuption in Section 3.1. However, the background gap is not always related to named entities as the year "1800" in the second example in Figure 1. This is annotated as explicitation since French readers are well acquainted with the period of the French revolution, while non-French readers may not be. Deciding whether a non-named entity, such as a certain period of time, needs explicitation and generating an explanation for it are more challenging problems than those related to named entities, and therefore, they require more advanced algorithms.

Our generation methods in Section 4.2 are not specifically tailored for the target audience. Instead, they are designed to retrieve information using the Wikidata and Wikipedia API in a simplified manner. In addition to the analysis of Section 6, the annotators point out some examples have too much information integrated into one place, for example, "Augustów, Sejny, **Poland**", and suggest the replacement rather than the mere addition like "Augustów, **Poland**" by removing unnecessary details. This failure case clearly displays the limitation of our generation methods, where the generated explanation is both incorrect and needlessly adding another difficult term in the explicitation for the target audience. Another example could be explaining a French-based entity, "J'accuse...!". The explanation from Wikipedia ("the open letter published by Emile Zola in response to the Dreyfus affair.") barely helps target reader as it introduces another unknown information like "Emile Zola" or "Dreyfus affair". When generating explicitation, a human translator would consider the importance of the time period and decide to do recursive explicitation which recursively explains the not well-known words in the explicitation itself. (e.g., "Dreyfus affair, *a political scandal in France, 1906*") These require further exploration of generation methods that need to be conditioned on target readers.

Our proposed method of automating explicitation is grounded in structured data. It enables precise control over deciding and generating explicitation by benefiting from consistency and enrichment of the data while avoiding hallucinations as Large Language Models (LLM). However, this approach may suffer from rigidity and limited creativity, thus having less flexibility on dealing with diverse natural language input, which could lower the quality

of explicitation. Exploring integration of structured data and language models could be next feasible step as in Yao et al. (2023).

## Ethics Statement

The workers who annotated and evaluated the explicitation candidates extracted from WikiMatrix and our automatically generated explicitation were paid at a rate of USD 0.25 per explicitation example. The resulting estimated hourly wage is above the minimum wage per hour in the United States.

## Acknowledgment

We thank the anonymous reviewers, Pranav Goel, Sathvik Nair, Ishani Mondal, Eleftheria Briakou and the members of the CLIP lab at UMD for their insightful and constructive feedback. This work was funded in part by NSF Grants IIS-1750695, IIS-2147292, and IIS-1822494.

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

---

**Algorithm 1:** Detecting explicitation span in bitext

---

**Input:** Sentence pair $(X_{src}, X_{tgt})$,
$\quad\quad E = \{e | \text{named entity } e \text{ in } X\}$
**Output:** Candidate $C$ with unaligned
$\quad\quad\quad$ segment $u$ and its related entity $e$
1 $U = \text{unaligned\_segments}_{tgt}(X_{src}, X_{tgt})$
2 $R = \{(e, u) | \text{is\_near\_and\_related}(e, u),$
$\quad\quad e \in E, u \in U\}$
3 $C = \{(e, u) | \text{decide\_explicitation}(e), \text{ \#Alg2}$
$\quad\quad (e, u) \in R\}$
**Return:** $C$

| Dataset | Guesser | Buzzer | Task | Expected Wins (EW) | Expected Wins with Oracle Buzzer (EWO) | Full Input Accuracy |
|---|---|---|---|---|---|---|
| XQB-*pl* | Rnn | Rnn | English QA | 0.35 | 0.70 | 0.62 |
| | LLaMA | Threshold | English QA | 0.26 | 0.64 | 0.62 |
| | LLaMA | Threshold | Polish QA | 0.17 | 0.56 | 0.43 |
| XQB-*es* | Rnn | Rnn | English QA | 0.30 | 0.65 | 0.48 |
| | LLaMA | Threshold | English QA | 0.10 | 0.72 | 0.50 |
| | LLaMA | Threshold | Spanish QA | 0.09 | 0.75 | 0.53 |

Table 6: Baseline performance of the original question without the explicitation. LLAMA is comparable or better without finetuning compared to RNN models trained on Quizbowl datasets.

```
Give your top guess with the confidence score to the following English questions:
Question: "This country has a winding, unpaved road that crosses the Los Yungas region and is dubbed as
"the most dangerous road in the world". A moutain that overshadows the city of Potosi in this country
provides a large part of the silver ore that made Spain rich during the colonial era. This country's
capital city is the highest in the world and shares Lake Titicaca with its northwestern neighbour, Peru.
For 10 points, name this landlocked country in South America which has two capital cities: Sucre and
Paz."
Top guess and its confidence score: ("Bolivia", 0.6)

Question: "This person appears as one of the main characters in the detective novel, "Teoría del
Manglar", written by Luis Carlos Musso, winner of the Miguel Riofrio National Literature Competition.
Yoshinori Yamamoto, revealed that he had managed to collect more than 4,500 recordings. He is one of
Ecuador's most popular singer-songwriters and his most famous song is "Nuestro Jurament..."
Top guess and its confidence score: ("Julio Jaramillo", 0.1)

Question: "A famous portrait of this man, created by Jacques-Louis David, shows him pointing to the sky
while he prepares to drink hemlock. The account of this man's execution was written by one of his
students in "Apology". For 10 points, name this Athenian philosopher who taught thinkers like Plato and
is famous for saying, I know only one thing: that I know nothing"."
Top guess and its confidence score:
```

Figure 5: Example prompt input to LLAMA for multilingual QA task. This example is from XQB-*es* and English QA task.

---

**Algorithm 2:** Deciding the necessity of explicitation on given entity

**Input:** Entity $e$
**Output:** True if Entity $e$ needs explicitation else False
**Param:** Language Pair $(l_{src}, l_{tgt})$, $P = \{(prop, \tau) | \text{Property and its threshold}\}$

1 **for** $(prop_k, \tau_k) \in P$ **do**
2    **if** *not* $(prop_k(e|l_{src}) - prop_k(e|l_{tgt}) > \tau_k)$ **then**
3      Return False
4 **end**
5 **if** *is_general(e)* **then**
6    Return False
   **Return:** True

---

| | | Explicitation with Answer | | Explicitation without Answer | |
|---|---|---|---|---|---|
| Type | Metric | Orig | Expl | Orig | Expl |
| SHORT | EW | 0.44 | 0.29 | 0.08 | 0.18 |
| | EWO | 0.82 | 0.84 | 0.51 | 0.54 |
| | FIA | 0.43 | 1.00 | 0.46 | 0.34 |
| MID | EW | 0.29 | 0.26 | 0.04 | 0.15 |
| | EWO | 0.66 | 0.71 | 0.48 | 0.50 |
| | FIA | 0.65 | 0.85 | 0.43 | 0.39 |
| LONG | EW | 0.27 | 0.20 | 0.05 | 0.06 |
| | EWO | 0.63 | 0.67 | 0.49 | 0.47 |
| | FIA | 0.64 | 0.68 | 0.41 | 0.30 |

Table 7: Performance differences by explicitation in the separate case of answer inclusion in XQB-*es*.

## A Baselines Comparison

Table 6 shows the baseline wins metric of original questions without explicitation. We also compare the English RNN models used in QANTA (Rodriguez et al., 2019) and SimQA (Han et al., 2022). RNN has higher EW as it uses the guesser and the buzzer specifically trained for Quizbowl. However, EWO and Full Input Accuracy are not affected by the buzzer, so LLAMA is comparable even though

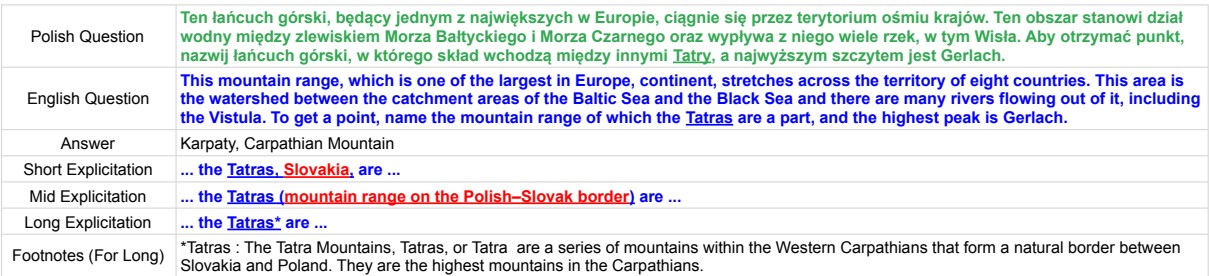

| | |
|---|---|
| Polish Question | **Ten łańcuch górski, będący jednym z największych w Europie, ciągnie się przez terytorium ośmiu krajów. Ten obszar stanowi dział wodny między zlewiskiem Morza Bałtyckiego i Morza Czarnego oraz wypływa z niego wiele rzek, w tym Wisła. Aby otrzymać punkt, nazwij łańcuch górski, w którego skład wchodzą między innymi Tatry, a najwyższym szczytem jest Gerlach.** |
| English Question | **This mountain range, which is one of the largest in Europe, continent, stretches across the territory of eight countries. This area is the watershed between the catchment areas of the Baltic Sea and the Black Sea and there are many rivers flowing out of it, including the Vistula. To get a point, name the mountain range of which the Tatras are a part, and the highest peak is Gerlach.** |
| Answer | Karpaty, Carpathian Mountain |
| Short Explicitation | **... the Tatras, Slovakia, are ...** |
| Mid Explicitation | **... the Tatras (mountain range on the Polish–Slovak border) are ...** |
| Long Explicitation | **... the Tatras* are ...** |
| Footnotes (For Long) | *Tatras : The Tatra Mountains, Tatras, or Tatra are a series of mountains within the Western Carpathians that form a natural border between Slovakia and Poland. They are the highest mountains in the Carpathians. |

Figure 6: Generation example from our experiment in extrinsic evaluation in XQB-*pl*.

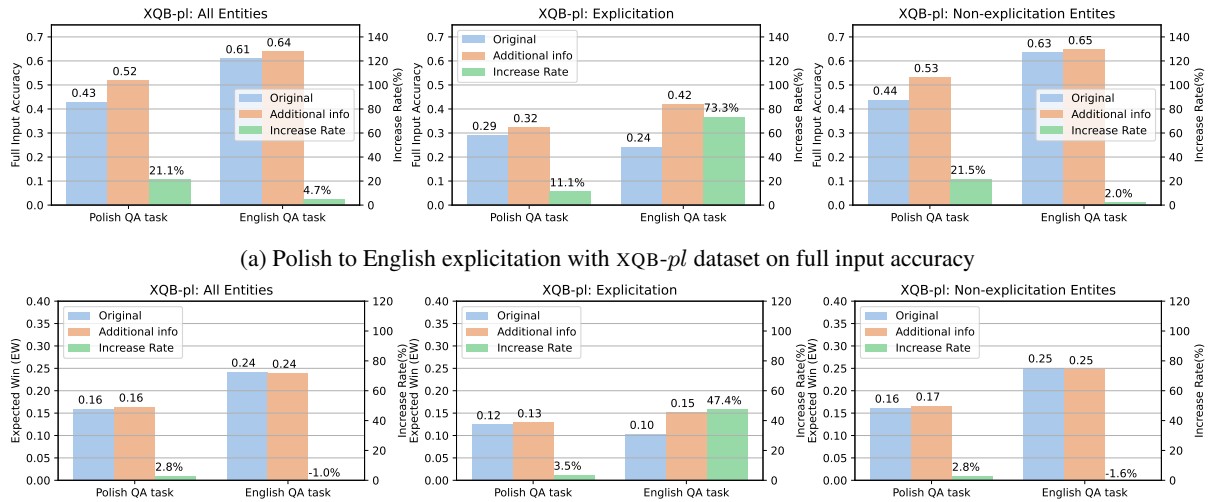

(a) Polish to English explicitation with XQB-*pl* dataset on full input accuracy

(b) Polish to English explicitation with XQB-*pl* dataset on EW

Figure 7: Comparison between the effect of additional information on all entities (left), on the entities chosen for explication (center), and the entities not chosen (right) on Polish to English XQB-*pl* with a metric of full input accuracy and EW with generation type of MID. Higher increase rate in English QA task in the center plot while the opposite trends in the right and left plots indicate our decision algorithm for automatic explicitation is valid.

it is not specifically fine-tuned on the task. While English and Spanish are comparable, there is an accuracy gap between English and Polish. This might suggest that the knowledge transfer between the languages may not happen effectively within a single multilingual model. Also, the gap between pl-en might be due to the difference in the amount of text seen in certain languages, as Polish has a smaller number of speakers compared to Spanish.

## B Data Collection Frameworks in Details

After extracting the explicitation candidates from the bitext corpus, we present the candidates to the human annotators and let them label whether the candidates are explicitation or not and the span of explicitation, and this section describes the details of annotation. The image of the annotation framework is in Figure 8a.

We present source and target sentences from the parallel corpus and mark unaligned segments in the target sentence as red while underlining the recognized named entities on both sides. We also provide the gloss in the target language with the Google Translate function for easy word-to-word comparison between source and target.

The first question we ask is to categorize the unaligned word. The annotators may select three classes of the role: 1) "Additional Information", 2) "Paraphrase (Equivalent and no additional info)" and 3) "Translation Error/Noise". Usually, the unaligned word is "Additional Information" where the contents do not exist in the source sentence (e.g. *la Sambre → the Sambre* **river**). However, it can be mistakenly selected as an unaligned word even though there is corresponding content on the source side, which falls into "Paraphrase". For example, *Pan de Antas → Antas breads* is the case for this as both phrases are equivalent. If the unalignment is due to a translation error or if the unaligned word is too divergent, we instruct the annotators

to select "Translation Noise/Error" (e.g. *participa* → *introduce*). We focus on the class "Additional Information" as it could possibly contain real explicitation which continues to the next question.

The second question is to decide "Is this explicitation". We ask annotators "Does this additive description explicitly explain the implicit general knowledge of the source language (SL) speaker for the target language (TL) speaking audience?". We indicate to them that this additional information is expected to be more useful to the target readers but not necessary to most speakers of the source language, and such implicit knowledge is less likely to be known by TL speaker compared to someone who is fluent in SL or familiar with SL speaking culture. This kind of general knowledge becomes explicit by the translator to enhance the reader's understanding of the translated text. For example, let's see *la Sambre → the Sambre **river***. This would be an example of explicitation, as it gives more context to the target audience who is not familiar with the name "Sambre" by adding the word "river" which does not exist in the source, while source language speakers may not need such explanation because it could be obvious to them. However, *Jeremy → **her policeman husband** Jeremy* is not explicitation but a simple addition of specific facts because the named entity is not famous figures or the added facts are not commonly well-known knowledge in the community of source language speakers.

If the annotator selects yes to the second question, then we instruct them to mark the span of explicitation in both the source and target sentence and leave a note about the reason for their decision if they have any as in Figure 8b.

## C  Types of Explicitation—Full Examples and Analysis

Table 8 shows additional examples of explicitation are categorized into five types from Table 2. One representative case of explicitation is introducing the hypernym of the entity ($h$) or adding titles/occupations to the human name ($o$). While the name representation alone might be clear for source language speakers due to its familiarity, the target audience may lack awareness or familiarity with the name.

Acronym Expansion ($a$) is also common types of explicitation (Baumgarten et al., 2008; Gumul et al., 2017). However, not all acronym expansion is marked as explicitation by annotators if the acronym is not commonly used by native speakers.

Full name representation is considered as one type of explicitation (Klaudy, 1996). A famous person's first or last name is often omitted for convenience. As our annotator comments, we do not add the first name, William when mentioning Shakespeare, and likewise, there's no need to add the first name, Miguel de to Cervantes ($f$–1) in Spain. Full name representations ($f$) are marked as explicitation as their name is famous within their country but presented in full name in English as it might not be the case outside.

The most common type is adding nationality information to the entity, especially for the location. Usually, the name of the nation is added to the name of a not well-known city, providing the context information of its country.

All these types can be integrated into one example ($i$). Typically, the explicitation adds the nationality and its title or the hypernym for the target audience who might not know who or what the entity is. The form of explicitation is diverse. The additional information could be accompanied by prepositions or integrated into the form of appositive with commas or in a parenthetical expression.

## D  Analysis on Answer Inclusion Cases

Table 7 shows the performance changes by explicitation in the separate case of answer inclusion. The original performance before the explicitation is already higher in the case of answer including explicitation compare to without answer. This indicates that the answer inclusion tends to happen in easy questions, where the original question text is already evident and the additional information is highly probable to include the answer because it is so obvious.

## E  Comparison between Explicitation and Non-explicitation

In Figure 7, we specifically examine the validity of the decision algorithm by comparing the influence of additional information in all named entities (left) and in non-explicitation (right) to those in the explicitation (center). Here, the non-explicitation indicates the entities with additional information that is not considered to be needed to do explicitation. In the explicitation results (center), the increase rate of the English QA task is higher than that of the Polish one, while additional information in all entities (left) and non-explicitation (right) is

| # | Type | Source | Target |
|---|---|---|---|
| (1) | Hypernym ($h$) | la Sambre | the Sambre **river** |
| (2) | | Belwederze | Belweder **Palace** |
| (3) | | Pleszowa | **village of** Pleszów |
| (1) | Occupation/ | Javier Gurruchaga | **showman** Javier Gurruchaga |
| (2) | Title ($o$) | Jana III | **king** John III **Sobieski** |
| (3) | | Juan Carlos de España | **Prince** Juan Carlos of Spain |
| (1) | Acronym | PP | **People 's Party** (PP) |
| (2) | Expansion ($a$) | TVE | **Televisión Española of Spain** (TVE) |
| (3) | | PCE | **Communist Party of Spain** |
| (4) | | PSL | **Polish People 's Party** |
| (1) | Full names ($f$) | Cervantes | **Miguel** de Cervantes |
| (2) | | Piłsudskiego | **Józef** Piłsudski |
| (3) | | Sierakowa | Sieraków **Wielkopolski** |
| (1) | Nationality ($n$) | Felipe II | Philip II **of SPAIN** |
| (2) | | Toruniu | Toruń (**Poland**) |
| (3) | | Cuenca | Cuenca **in Spain** |
| (4) | | Troyes | Troyes **, France** |
| (1) | Integrated ($i$) | Dominique de Villepin | **former French Prime Minister** Dominique de Villepin |
| (2) | | El País | **Spanish newspaper** El País |
| (3) | | Zygmunt III Waza | **Polish King** Sigismund III Vasa |
| (4) | | Przekrój | **Polish weekly magazine** Przekrój |

Table 8: Additional examples of explicitation from Table 2. We categorize the collected explicitation example into five types. Integrated is for the examples that include two or more types of explicitation. The boldface indicates added text by explicitation. The spans of added text are marked by annotators. Full-sentence version available in Figure 8b.

more effective in Polish QA task in both full input accuracy and EW metrics. This demonstrates the effectiveness of our algorithm in determining the need for explicitation.

## F Additional Related Work

**Studies on *pragmatic* explicitation.** There are several succeeding research that studies on *pragmatic* explicitation (Saldanha, 2008; Becher, 2010; Adil Abdulwahab, 2012). Adil Abdulwahab (2012) emphasizes the need for *pragmatic* explicitation in the translation of English short stories into Arabic. Becher (2010) did a rigorous search of every type of explicitation in a corpus of English popular scientific magazine articles and their German translations and find that *pragmatic* explicitation is rare, which corresponds to our empirical results in Section 3.6.

**Cross-cultural NLP and Wikipedia.** Despite the impressive gains in NLP fields, Hershcovich et al. (2022) identifies the intractable challenges in cross-cultural NLP by pointing out that the production

and the consumption of the contents are largely varied not just by language but also by culture. An epitomic example would be the multilinguality in Wikipedia and its differences in content across languages. Callahan and Herring (2011) specify a cultural bias in the content, especially if it is related to famous people. Massa and Scrinzi (2012) emphasize differences in perspectives among diverse Wikipedia communities in distinct languages. Hale (2014) and Hecht and Gergle (2010) find a "surprisingly" small degree of content overlap between different languages in Wikipedia. Still, various efforts have been made to reflect these cultural varieties in the data and the models (Dev et al., 2023). As one of the various efforts to accommodate cultural diversity better to serve users of cross-cultural NLP systems, we explore the possibility of automatic explicitation to be beneficial on a larger scale which could help bridge the cultural gap between the language user communities.

| | The Named Entity are recognized and underlined. Unaligned English words near the Named Entity will be highlited as Red. | Please classify what is the closest role of Red words in English when you compare the sentence between a source and target | If "Additional Information", Is it Explicitation? (more details in note) [1] | If it is explicitation, Please specify the span with "{bracket}" that explicitation happens in the source and the target. | Any comments (why do you think this is explicitation, or why do you think this is useful or not useful) |
|---|---|---|---|---|---|
| Underline Red | The Named Entity are recognized and underlined. Unaligned English words near the Named Entity will be highlited as Red. | | | | |
| Source | Les travaux commencent sur une ligne d' ouvrages de campagne en face de la forteresse , avec ses points de terminaison sur la Sambre, pour couper la forteresse de l' accès par voie terrestre . | | ☐ | Span annotation is incomplete. Please check if there is missing pairs or unmatched number of pairs between source and target | |
| Google Tran | The work begins on a line of country works in front of the fortress, with its termination points on the Sambre, to cut the fortress of land access. | | | | |
| Target | Work was begun on a line of field works that ran in front of the fortress , with its endpoints on the Sambre river , cutting the fortress off from overland access . | | | | |

(a) Annotation framework for collecting explicitation example.

| | | |
|---|---|---|
| h-(2) | Source | Młodość spędziła głównie w Warszawie , gdzie mieszkała z rodziną w {Belwederze} oraz w Sulejówku w dworku „ Milusin " podarowanym Piłsudskiemu przez żołnierzy . |
| | Google Translate | She spent her youth mainly in Warsaw, where she lived with her family in {Belvedere} and in Sulejówek in the manor of "Milusin" donated to Piłsudski by soldiers. |
| | Target | She spent her youth mainly in Warsaw , living with her family at the {Belweder Palace} , and in Sulejówek at the cottage of Milusin , which Piłsudski had received as a gift from his soldiers . |
| | Annotator's comment | It gives more context to the target audience who is not familiar with the name "Belweder" by adding word Palace which does not exist in the source, while source language speakers may not need such explanation because it could be obvious to them. |
| o-(2) | Source | Obok prezentacji bogatych zbiorów sztuki europejskiej i dalekowschodniej , część centralną pałacu poświęcono pamięci {Jana III} i wspaniałej przeszłości narodowej . |
| | Google Translate | In addition to the presentation of the rich collections of European and Far Eastern art, the central part of the palace was devoted to the memory of {Jan III} and the great national past. |
| | Target | Besides European and Oriental art , the central part of the palace displayed a commemoration of {king John III Sobieski} and the glorious national past . |
| | Annotator's comment | Very clear example. 'Jan III' is enough for a Polish person to know it's about Sobieski and the he was a king. |
| a-(2) | Source | La única cadena de televisión que pudo grabar en el interior de la plaza en la noche del 3 al 4 de junio fue {TVE} . |
| | Google Translate | The only television network that could record inside the square at night from June 3 to 4 was {tve}. |
| | Target | The only network which was able to record shots during the night of 4 June was {Televisión Española of Spain ( TVE )} . |
| | Annotator's comment | In this case it's a clear explicitation and it's useful because in the source text there is an acronym that everybody in Spain would easily identify, but it's not obvious for non native speakers. |
| a-(4) | Source | W dniu 28 czerwca 1914 zwrócił się wraz z Franciszkiem Wójcikiem z odezwą do chłopów o finansowe popieranie {PSL – Lewica} . |
| | Google Translate | On June 28, 1914, he and Franciszek Wójcik with a appeal to the peasants for financial support {PSL - left}. |
| | Target | On June 28 , 1914 , together with Franciszek Wójcik , he addressed the peasants with financial support for the {Polish People 's Party " Left} " . |
| | Annotator's comment | PSL is a commonly known acronym and is translated to Polish People's Party - this acronym is frequently used in the source language community and isn't familiar to the target language speaking community. Expanding it will help the target reader's understanding, it counts as explicitation. |
| f-(1) | Source | Ellas les regalan un libro , recordando el fallecimiento de dos grandes de la literatura europea , {Cervantes } y Shakespeare y del hispanoamericano Inca Garcilaso . |
| | Google Translate | They give them a book, remembering the death of two greats of European literature, {Cervantes} and Shakespeare and the Spanish -American Inca Garcilaso. |
| | Target | The women give the men a book , remembering the death and burial respectively of two great European literary personalities , {Miguel de Cervantes} and Shakespeare , and the Spanish notable literary personality , Inca Garcilaso . |
| | Annotator's comment | It's just adding the name of a really famous person, so there's no need as this person is know by his surname. The same way that they didn't add the first name (William) when mentioned Shakespeare, there's no need to add the first name (Miguel de) to Cervantes. |
| n-(1) | Source | En el siglo XVI y despúes de la batalla de San Quintín que acabó el 10 de agosto de 1557 , Fiesta de San Lorenzo , {Felipe II } decidió construir { San Lorenzo del Escorial } en honor al santo . |
| | Google Translate | In the 16th century and after the battle of San Quintín that ended on August 10, 1557, feast of San Lorenzo, {Felipe II} decided to build {San Lorenzo del Escorial} in honor of the saint. |
| | Target | In the 16th century and after the Battle of Saint Quentin that ended on Saint Laurent 's day , AUGUST 10th 1557 , { Philip II of SPAIN } built the {Palace of San Lorenzo del Escorial }, near Madrid . |
| | Annotator's comment | This is very useful, because for me as a native speaker it's obvious what is the San Lorenzo del Escorial and who Felipe II was, but non native speakers could straggle with that. |
| i-(1) | Source | Dans cette promotion fut également admis , aux côtés de Patrick Levaye , Richard Descoings ( ancien directeur de l' Institut d' études politiques de Paris ) , Patrick Galouzeau de Villepin ( {frère de Dominique de Villepin} ) , Jean-François Cirelli ( président de BlackRock France ) , Jean-Claude Mallet ( ancien secrétaire général de la défense nationale ) , François Asselineau ( deuxième de promotion et Président de l' Union Populaire Républicaine ) , entre autres . |
| | Google Translate | In this promotion was also admitted, alongside Patrick Levaye, Richard Descoings (former director of the Institute of Political Studies in Paris), Patrick Galouzeau de Villepin ((brother of Dominique de Villepin)), Jean-François Cirelli (President From Blackrock France), Jean-Claude Mallet (former secretary general of national defense), François Asselineau (second of promotion and president of the People's Republican Union), among others. |
| | Target | In this promotion was also admitted , at the sides of Patrick Levaye , Richard Descoings ( director of the Paris Institute of Political Studies ) , Patrick Galouzeau de Villepin ( {brother of the former French Prime Minister Dominique de Villepin} ) , François Peny ( General secretary of préfecture of the Gironde ) , Jean-François Cirelli ( president of Gaz de France ) , Jean-Claude Mallet ( former General secretary of French National defense ) , so on . |
| | Annotator's comment | This is a good example of explicitation, providing the target audience with the contextual information about Dominique de Villepin, even though he is a well-known figure in France and requires no further explanation for the French audience. |

(b) Full sentence version of Table 2. Brackets are the span of explicitation marked by annotators including original entities and added information while red fonts are unaligned ones. Underlines in Source and Target is named entities marked by NER model.

Figure 8: Annotation Framework and full sentence example of explicitation with annotator's comments.