# OpenReview forum: "Bridging Background Knowledge Gaps in Translation with Automatic Explicitation"
_EMNLP/2023/Conference — EMNLP 2023 Main_

### Official Review · Reviewer_CM4F · 2023-08-02

**Soundness:** 4

**Excitement:**

3: Ambivalent: It has merits (e.g., it reports state-of-the-art results, the idea is nice), but there are key weaknesses (e.g., it describes incremental work), and it can significantly benefit from another round of revision. However, I won't object to accepting it if my co-reviewers champion it.

**Paper Topic And Main Contributions:**

This paper is about incorporating explicitations in translation to bridge the gaps due to the socio-cultural background between people speaking the source and target language. For instance, a well-known French politician is less likely to be known to an English-speaking audience, thus requiring additional information in the French-to-English translated text. This paper contributes twofold: they release a new dataset, WikiExpl, which is a dataset of entities described differently across languages, and then focus on explicitation by exploring whether making implicit knowledge more explicit helps downstream tasks such as multilingual question answering.

**Questions For The Authors:**

A. On line 214, the authors introduced a threshold. How is it determined? How does it change between pairs of source and target languages?

B. On line 219, the authors chose to exclude entities "well known globally". Why? Maybe it can be shown that by including these entities the performance downgrades. And how can you define algorithmically what a "well-known" entity is? Even though these entities appear in 250+ Wikipedia pages this does not mean that a reader should be aware of its existence.

C. On line 148, the authors say "We choose a not-too-clean parallel corpus from Wikipedia". Why? How do you choose such corpora?

D. On line 258, the authors say: "For proximity, we decide that a segment is near and entity if it is within three words distance". Why? Did they try different distances and witness a meaningful drop in performance? This study should be integrated.

E. On line 263, the authors say: "For the decision algorithm, we set the threshold as 1 for the property of the number of hops, which "usually" selects the entities that have direct relational links to the source country". Like for question 4, why? Maybe with more hops, you can find an entity that is known to multiple countries.

**Reasons To Accept:**

1. They produce a carefully generated and human-annotated dataset including qualitative and quantitative analyses.

2. They perform extensive intrinsic (with human annotators) and extrinsic evaluation of the application of explicitation on the QuiBowl dataset (both Spanish-English and Polish-English), showing also an increase in performance following the metrics suited for the aforementioned dataset.

3. Human translators include a variety of explicitation when translating, but current machine translation models do not. This study sheds light on this previously neglected, but useful type of information, utilizing a scientific approach to the problem.

**Reasons To Reject:**

1. The dataset, as described in Section 3, seems very small. The motivation behind this statement is that the authors write: "About 1-3% of sentences are explicitation candidates" (row 284, subsection 3.6) and "The overall ratio of the final explicitation example from the initial corpus is 0.1 - 0.3%" (row 290, subsection 3.6). This seems also to be confirmed from Table 1, where the number of explicitations is 67, 44, and 36 for Polish, Spanish, and French respectively.

2. The motivations behind specific choices concerning the dataset and some hyperparameters are not well documented and explained, for instance:
    - 2.1 On line 214: the threshold introduced.
    - 2.2 On line 219: the choice behind the fact that they exclude entities "well known globally".
    - 2.3 On line 148: "We choose a not-too-clean parallel corpus from Wikipedia"
    - 2.4 On line 258: "For proximity, we decide that a segment is near and entity if it is within three words distance".
    - 2.5 On line 263: "For the decision algorithm, we set the threshold as 1 for the property of the number of hops, which usually selects the entities that have direct relational links to the source country".

**Reproducibility:**

3: Could reproduce the results with some difficulty. The settings of parameters are underspecified or subjectively determined; the training/evaluation data are not widely available.

**Reviewer Confidence:**

3: Pretty sure, but there's a chance I missed something. Although I have a good feel for this area in general, I did not carefully check the paper's details, e.g., the math, experimental design, or novelty.

**Typos Grammar Style And Presentation Improvements:**

Table 1, the "Explicitation-French" cell has "36^5"written, but there is no footnote 5, I guess this is a typo.

---

> ### Author Rebuttal · Authors · 2023-08-29
>
> R-1) “The dataset, as described in Section 3, seems very small.”
> →  While the dataset may seem small, particularly when compared to the wealth of massively multilingual benchmarks that annotate relatively common language phenomena, we argue that it provides a sufficient basis for a first study of cross-lingual explicitations, a relatively infrequent and understudied phenomenon. Furthermore, we contend that the methodology we introduced can be used to expand it further.
>
> R-2) “The motivations behind specific choices concerning the dataset and some hyperparameters are not well documented and explained,”
> → We will improve the description of the candidate extraction and of the dataset construction as suggested. Thank you!
>
> 2-A) “On line 214, the authors introduced a threshold. How is it determined? How does it change between pairs of source and target languages?”
> →  The thresholds are set on a development set drawn from a preliminary annotation stage, which uses the framework described in the paper but with non-expert annotators (volunteer graduate and undergraduate students). The method is applied to each language and it results in the same hyperparameter values.
>
> 2-B) ”On line 219, the authors chose to exclude entities "well known globally". Why? Maybe it can be shown that by including these entities the performance downgrades. And how can you define algorithmically what a "well-known" entity is? Even though these entities appear in 250+ Wikipedia pages this does not mean that a reader should be aware of its existence.”
> → We anticipate that entities that are well-known globally are less likely to require explicitation, thus we exclude them from our annotation effort.
> We use the number of languages in which a Wikipedia page is available for this entity to quantify how well-known it is. We recognize that this is a simplification, but consider that this is a reasonable assumption in the context of Wikipedia.
>
> 2-C) ”On line 148, the authors say "We choose a not-too-clean parallel corpus from Wikipedia". Why? How do you choose such corpora?”
> →  If the parallel corpus is too “clean” or too parallel, it is more likely to contain literal translations rather than explicitation examples. Explicitation is by definition a non-literal translation, making an implicit detail in the source explicit in the target, and thus introducing unaligned content in one of the languages.
>
> 2-D) “On line 258, the authors say: "For proximity, we decide that a segment is near and entity if it is within three words distance". Why? Did they try different distances and witness a meaningful drop in performance? This study should be integrated.”
> → We determine the distance of three based on the qualitative analysis of the sample from the preliminary annotation stage mentioned above 2-A) and the efficiency of the candidate extraction process.
> We define a distance as a difference in the index of the tokenized words. The distance between an entity and the closest token of its explicitation segment is mostly one to two in our main target language, English. In the example of “Sambre” in Table 3, the distance can be one in the type Short, two in Mid (comma is separated by tokenizer), and three in Long (a bracket and an article are in between). The optimal distance may change according to the target language.
> On the practical side, the distance of three is the most efficient in terms of extracting the candidates, as we use APIs to determine the necessity of explicitation (decision). Increment of the distance threshold in proximity from three to four would have slightly better recall for the candidates, however, not practical enough to cover the increased computational cost.
>
> 2-E) “ On line 263, the authors say: "For the decision algorithm, we set the threshold as 1 for the property of the number of hops, which "usually" selects the entities that have direct relational links to the source country". Like for question 4, why? Maybe with more hops, you can find an entity that is known to multiple countries.”
> → The main motivation for using the number of hops between the entity and certain language-speaking countries in the relational graph is to see the cultural boundedness of an entity to the source language-speaking country and unboundedness to the target country.
> In our trial with n >= 1 hops and |\tau| >= 1 on the sample from the preliminary annotation stage mentioned above 2-A), we find that n > 1 hops lack the boundness while having quadratic increase of computational cost, and |\tau| = 1 is enough for the boundness gap when we set the maximum number of hops to one. The parameter of |\tau| > 1 will reinforce the shift from source to target and thus give more confidence in the decision of explicitation, however, also increase the computational burden. Therefore, we decided to set this property with n=1 hop and \tau = -1 where it becomes checking whether the entity has a direct link to the source side and does not have it to the target side.
>
> Typo) “Table 1, the "Explicitation-French" cell has "36^5"written, but there is no footnote 5”
> → The actual footnote text was “only 150 are annotated.”  in Table 1 for the “French” column and “Explicitation” row, where the footnote command in a table does not appear in the paper. We will correct this error in our future draft.

---

### Official Review · Reviewer_vd7U · 2023-08-09

**Soundness:** 4

**Excitement:**

4: Strong: This paper deepens the understanding of some phenomenon or lowers the barriers to an existing research direction.

**Missing References:**

Ralph Krüger wrote a few papers on explicitation in MT which I was surprised to not see. This work is more focused on what explicitation already appears in existing MT systems but it seems like a relevant perspective.
Krüger, R. (2020). Explicitation in neural machine translation. Across Languages and Cultures, 21(2), 195-216.

**Paper Topic And Main Contributions:**

This paper introduces the apparently novel task of adding semantic explicitations to automatic translations. A dataset of such explicitations is also built and made available by the authors, which involves detecting candidate explicitations in parallel text and selecting/annotating them. Various assumptions are made in the automated selection steps before manual annotation.

In the generation task, the explicitations are generated from knowledge bases rather than from generative language models, and they are evaluated using two tasks: an intrinsic evaluation task where the annotators evaluate them in a categorial scheme, and question answering as an extrinsic evaluation task. Different types of explicitation are distinguished. The procedure is at least partially grounded in the translation studies literature and helps to address a weakness of of MT systems that rely too much on optimizing BLEU.

**Reasons To Accept:**

The work introduces an interesting task that can help improve translation quality, but that has probably been ignored because it does not increase scores on the typical metrics. The dataset should be useful in enabling others to do similar work. The contributions are substantial, with both a dataset, new task and two evaluations, involving manual annotation at various steps. It should be of fairly broad interest to the MT community.

**Reasons To Reject:**

My main concern is that there are a lot of assumptions about explicitations built into the work, which might narrow the definition of explicitations considerably. For example, that explicitation always pertains to explicitly and nearby mentioned entities. The authors also mention a few in the limitations, such as the need to have a link to a single country. This also cannot be easily addressed in followup work, as many of these assumptions were built into the data collection. Some further assumptions also exist in the modeling, such as what type of data to fetch from where. These assumptions also make the work fairly specific to English and to certain resources, but can be changed more easily. It is clear that some assumptions have to be made, but these were not really examined at least not in the paper.

To address this, it would have been interesting for example to have an (expert?) annotator go over a text manually, see where explicitations would be helpful (without strictly adhering to the assumptions about explicitation made by the authors) and then count how many of those are actually provided in its translation. This would help estimate recall.

**Reproducibility:**

4: Could mostly reproduce the results, but there may be some variation because of sample variance or minor variations in their interpretation of the protocol or method.

**Reviewer Confidence:**

2: Willing to defend my evaluation, but it is fairly likely that I missed some details, didn't understand some central points, or can't be sure about the novelty of the work.

**Typos Grammar Style And Presentation Improvements:**

I tried to find out information about the annotators and annotation but it is a bit scattered throughout the paper. It would be better if it was introduced in a single place. We can get some glimpses of the procedure in the appendix but it would have been nice to see the full instructions!

There are capitalization issues in the references.

---

> ### Author Rebuttal · Authors · 2023-08-29
>
> R1) “there are a lot of assumptions about explicitations built into the work, which might narrow the definition of explicitations considerably.”
> →  It is true that we focus on pragmatic explicitation (Sec2, L118) that is one of the explicitation categories by Klaudy (1993, 1996, 1998). However, we think it is worth spotting light on even though it does not cover the all explicitation cases. Our primary aim is to enhance the understanding between the two language speakers transcending mere literal translation by addressing gaps in their background knowledge. This aligns seamlessly with our emphasis on explicitation.
> The three assumptions are based on the well-examined hypothesis of explicitation (Sec3.1, L152) by Blum-Kulka (1986) and the characteristics that define pragmatic explicitation, which make these “assumptions”' natural and reasonable to make:
> First assumption that explicitations are part of unaligned token sequences  (Sec 3.1, L157) comes with the very definition of the explicitation in general, a non-literal translation making an implicit detail in the source explicit in the target (Vinay and Darbelnet, 1958), which can be applied to most kinds of explicitation including pragmatic explicitation. Finding alignment discrepancies is commonly used to detect explicitation like in Lapshinova-Koltunski and Hardmeier (2017).
> Second and third assumptions that explicitations are close to culturally bounded named entities (Sec 3.1, L163-178) stem from and are grounded in the definition of pragmatic explicitation, In this context, a named entity epitomizes the prevailing form that bridges the gap between shared knowledge among communities.
> However, these assumptions mainly deals with addition of explanation of named entity rather than with phrase of expression like idioms or replacement which is also another manifestation form of the explicitation (e.g. Halloween → Ramadan in Adil Abdulwahab, 2012).
>
> R2) “To address this, Let annotators go over the text manually and see where explicitations would be helpful without assumptions and narrow definition. Then, count how many of those are actually provided in its translation, which could help estimate the recall.”
> → The suggestions would be a great idea to check if the pragmatic explicitation would be naturally occurring and really helpful, and this would be an alternative way to evaluate the decision and generation algorithm. A simple approach to implement this would be leveraging existing human translation containing explicitation. We are interested in delving into this direction as part of our future endeavors.
>
> Missing Ref) Ralph Krüge’s work
> → The analysis of automatic explicitation by the MT system seems closely relevant to our paper, and it would be interesting to see how the explicitation that is generated by the MT system is different from the pragmatic explicitation that is proactively generated by our algorithm. We will add further comparison analysis in our future drafts.
>
> Presentation Improvements)  information about the annotators and annotation but it is a bit scattered throughout the paper.
> → In our upcoming drafts, we will present the dispersed information across Sec 3.4 to Sec 3.6 and Appendix B in a more coherent manner within a single paragraph in Section 3.4.

---

### Official Review · Reviewer_SwKe · 2023-08-10

**Typos Grammar Style And Presentation Improvements:** N/A
**Soundness:** 4

**Excitement:**

4: Strong: This paper deepens the understanding of some phenomenon or lowers the barriers to an existing research direction.

**Missing References:**

None

**Paper Topic And Main Contributions:**

This paper delves into the concept of explicitation in translations, where the goal is to go beyond literal meanings to bridge cultural gaps and improve understanding between source and target languages. Professional translators use explicitations to provide context and background knowledge to the audience of the target language. Despite its potential to enhance translation quality, NLP research on explicitation has been limited due to a lack of proper evaluation methods. This paper introduces techniques for automatically generating explicitations, supported by the creation of the WIKIEXPL dataset. The resulting explicitations are assessed for their utility in enhancing multilingual question answering systems.

**Questions For The Authors:**

Could you elaborate on specific instances where literal translations fail to convey the intended meaning due to cultural or contextual differences?
The paper mentions previous research on explicitation in discourse translation. Can you highlight the differences between your proposed approach and these previous studies?
Please explain the process of creating the WIKIEXPL dataset in more detail. How did you choose the entities for annotation?
What were the criteria for selecting examples that require explicitation?

**Reasons To Accept:**

The paper effectively addresses the problem of translation not only as a literal process but as a means of conveying cultural nuances and background information to ensure better understanding. The creation of the WIKIEXPL dataset, annotated with human translators, is commendable. It serves as a valuable resource for training and evaluating the proposed explicitation techniques. Strengthening the evaluation and providing a clearer link to existing research will further enhance the paper's impact.
The paper demonstrates the practical relevance of explicitation by evaluating its impact on multilingual question answering, a real-world use case where understanding cultural context is crucial.


**Reasons To Reject:**

While the paper identifies the gap in research related to explicitation, it could further emphasize why current NLP approaches might struggle with it and how the proposed techniques address these limitations.

**Reproducibility:**

4: Could mostly reproduce the results, but there may be some variation because of sample variance or minor variations in their interpretation of the protocol or method.

**Reviewer Confidence:**

3: Pretty sure, but there's a chance I missed something. Although I have a good feel for this area in general, I did not carefully check the paper's details, e.g., the math, experimental design, or novelty.

---

> ### Author Rebuttal · Authors · 2023-08-29
>
> Q1) “Could you elaborate on specific instances where literal translations fail to convey the intended meaning due to cultural or contextual differences?”
> → Beside the example in Figure 1, Table 2 and 9, we can look at this example: "Nombreux sont ceux qui, devant une injustice, ont écrit leur « J’accuse… ! »." in French where the literal English translation is "Many are those who, in the face of an injustice, have written their ‘I accuse!’."
> Here the meaning of ``J'accuse...!'' in French goes beyond its literal meaning, and thus the “I accuse” translation in English is not appropriate. A human translator may add an explanatory description of “J’accuse… ! ” such as as “a famous open letter by a French novelist accusing the government in response to a miscarriage of justice.'' in the English translation sentence or in a footnote.
>
> Another example would be "Pozostałe 4 tys. ludzi zostało przewiezione na pobliski Majdanek." in Polish where the literal English translation is "The remaining 4,000 people were transported to the nearby Majdanek.". Polish speakers would understand that the reference to Majdanek is not simply to location name but that it was specifically a concentration camp.  Thus a human translator might translate the input as “The other 4,000 people were sent to the nearby KL Lublin / Majdanek concentration camp.”
>
> Q2) “The paper mentions previous research on explicitation in discourse translation. Can you highlight the differences between your proposed approach and these previous studies?”
> → Prior work has focused on explicitation of discourse markers such as discourse connectives, while we focus on explicitation that arises due to background knowledge gaps between language communities.
>
> For instance, Hoek et al. (2015) investigate whether expectedness influences the degree of implicitation and explicitation of discourse relations. Their approach is to first extract the parallel text that contains "because", "although", and "if" in English, Dutch and German. Then, they manually annotate the explicitness of the connective's translation in seven categories.
>
> Lapshinova-Koltunski and Hardmeier (2017) introduce an automatic method to extract examples of cross-linguistically divergent discourse structures from a corpus of parallel text, and quantitatively describe alignment discrepancies between English-German discourse-related phenomena from a language contrastive perspective.
> They define discourse-related structures as potential elements of coreference chains that can be either personal (e.g. she) or demonstrative pronouns (e.g. this). Their approach involves tools of part-of-speech tagging and dependency parsing to spot the discourse structures of interest ("DET","PRON"). With tagger and parser information, they extract parallel patterns with an alignment tool, and count the occurrences of the alignment discrepancy patterns (e.g. DET-nsubj → PRON-dobj). There are no manual annotations involved.
>
> Q3) “Please explain the process of creating the WIKIEXPL dataset in more detail. How did you choose the entities for annotation? What were the criteria for selecting examples that require explicitation?”
> → The process of creating the WIKIEXPL dataset starts with extracting the candidates based on the traits of the pragmatic explicitation described in Sec 3.1.
> First, we take the WikiMatrix bitext and do the detection of explicitation (Sec3.2, Algorithm 1): We run alignment tools to find unaligned models and NER tools to find entities in the bitext. Next, we pair the unaligned segments and entities if two are within three words distance and relevant by checking if the unaligned segment is included in the entity's Wiki page or fetched explanation from Wikidata.
> Second, we decide if the entity needs explicitation (Sec3.3, Algorithm 2): given the entity and the source/target language and knowledge graph, we see if a number of hops from the entity to source language speaking country is one and more than one for target language. Additionally, we consider globally-known entities that do not need explicitation, and exclude it from the candidate if it has more than 250 languages available for the Wikipedia page. Therefore, the criteria for selecting samples that require explicitation is cultural boundedness and global popularity.
> Now we have extracted candidates, we show them to a translator to do the annotations (Sec 3.4). We ask them label 1) whether the candidates are explicitation and 2) the span of explicitation. If two or more annotators annotate the candidate as explicitation, then we finally label it as explicitation (statistics in Table 1).

---

### Official Review · Reviewer_5DDw · 2023-08-12

**Soundness:** 3

**Ethical Concerns:**

Yes

**Excitement:**

2: Mediocre: This paper makes marginal contributions (vs non-contemporaneous work), so I would rather not see it in the conference.

**Paper Topic And Main Contributions:**

Inspired by WIKIEXPL, the work introduces several techniques for automatically generating explications, which is significant in achieving better accuracy in a multilingual question-answering framework.

**Reasons To Accept:**

The paper addresses a significant translation problem by automatically generating explications, and several improvement techniques have been designed.

**Reasons To Reject:**

The innovation, illustration, and writing need further improvement, and it is necessary to provide an introduction to the recent work in the background. Moreover,  they should conduct more comprehensive experiments to prove the algorithm's performance and make the experimental results more convincing.

**Reproducibility:**

3: Could reproduce the results with some difficulty. The settings of parameters are underspecified or subjectively determined; the training/evaluation data are not widely available.

**Reviewer Confidence:**

3: Pretty sure, but there's a chance I missed something. Although I have a good feel for this area in general, I did not carefully check the paper's details, e.g., the math, experimental design, or novelty.

---

> ### Author Rebuttal · Authors · 2023-08-29
>
> Thank you for your valuable feedback. We will improve the quality of our work by including recent works and clearer writing. We believe that the results presented support our main findings, but we welcome more specific suggestions in making our experiments more comprehensive.

---

### Official Review · Reviewer_Se4s · 2023-08-12

**Soundness:** 4

**Excitement:**

4: Strong: This paper deepens the understanding of some phenomenon or lowers the barriers to an existing research direction.

**Paper Topic And Main Contributions:**

This work introduces techniques for automatically generating pragmatic explicitations, to capture the different assumed knowledge of different languages. They create a collection of entities described differently across languages, referred to as WIKIEXPL, collected from Wikipedia. This dataset is used to identify entities that should be explained in translation. The extraction  is tested through an automatic evaluation with a multilingual question answering (QA) system, and intrinsically by human annotators.

**Reasons To Accept:**

* Well written and easy to follow
* Develop a process to detect pragmatic explicitation from Wikipedia parallel corpus, propose simple interesting ideas of using wikipedia entries/ distance for detecting
* Unique problem space
* Show positive results in intrinsic and automatic eval

**Reasons To Reject:**

* Given the very low instances of explicitation, the smaller size of the data and small number of language pairs make for weaker inferences on experimental performance. While the pol-english QA shows improvement, the spanish-english doesn’t on full input accuracy - EW/EWO and full input accuracy isn’t explained in the paper, which seems like a major omission since the performance claim is based on it.
* Both a pro and con the explicitation is tied to structured fields from wikipedia.



**Reproducibility:**

3: Could reproduce the results with some difficulty. The settings of parameters are underspecified or subjectively determined; the training/evaluation data are not widely available.

**Reviewer Confidence:**

3: Pretty sure, but there's a chance I missed something. Although I have a good feel for this area in general, I did not carefully check the paper's details, e.g., the math, experimental design, or novelty.

---

> ### Author Rebuttal · Authors · 2023-08-29
>
> R1) “Given the very low instances of explicitation, the smaller size of the data and small number of language pairs make for weaker inferences on experimental performance.”
> → While the dataset may seem small, particularly when compared to the wealth of massively multilingual benchmarks that annotate relatively common language phenomena, we argue that it provides a sufficient basis for a first study of cross-lingual explicitations, a relatively infrequent and understudied phenomenon. Furthermore, we contend that the methodology we introduced can be used to expand it further.
>
> R2) “While the pol-english QA shows improvement, the spanish-english doesn’t on full input accuracy”
> → Taken together, the results presented in Figure 3 support the claim that automatic explicitation is useful. We first explain the take-aways from the plot, before explaining the trends observed for spanish-english full inputs and the EW results.
>
> [plot explanation] In Figure 3-a, Full Input Accuracy (third column) shows a greater increase in English QA task (73% increase rate from 0.24 to 0.42) than Polish QA task (11% increase rate from 0.29 to 0.32) in XQB-pl with Explicitation.
> Higher increase rate of performance with additional explanation (Original vs Explicitation in the plot) in English QA task than in the source language QA task indicates the effectiveness of our automatic explicitation, as this gap of increase rate suggests that the added information is more helpful in English QA task and thus explicitation is adequate (Figure 2). XQB-pl (Fig 3-a) on Full Input Accuracy shows improvement from 11% in Polish QA task to 73% in English QA task which we can tell that our automatic explicitation is adequate in XQB-pl.
> In Figure 3-b, Full Input Accuracy shows no increase by explicitation on English QA task (0% increase rate from 0.47 to 0.47) and so does on Spanish QA task (2.2% increase rate from 0.40 to 0.41) in XQB-es with Explicitation. As mentioned, there is no improvement by explicitation in English QA task (0%) and even smaller than that in Spanish (2.2%).
>
> [Answer] → We attribute the different trends on Full Input accuracy for XQB-pl and XQB-es to different levels of difficulty of questions: the questions that includes culturally bounded entity in XQB-es are easier than in XQB-pl. If a question is already easy, then additional information would not be very helpful and thus shows a small or no increase rate like in XQB-es. This is corroborated by the huge accuracy gap of 0.20 between XQB-es and XQB-pl.
>
> The trend is different in Expected Win (EW). Both in XQB-pl and XQB-es show great improvements of increase rate from English QA task to Polish/Spanish QA task on EW. Evaluating models with the Expected Win (EW) metrics shows the benefits of automatic explicitation on a harder task than the Full Input Accuracy settings. The EW metric captures whether a player (including the QA system) answers questions both quickly and correctly.  Answering the question early and correctly is therefore harder than answering a question correctly given the full input.
>
>
> R3) “EW/EWO and full input accuracy isn’t explained in the paper,”
> → We will edit the paper to fully define the metrics rather than simply cite the references. We will explain that we adopt metrics from Han et al. (2022) and Rodriguez et al. (2019) (Sec 5, L461).  The Quizbowl QA system processes question text incrementally. At each step, it outputs the current guess and its decision of whether to offer its guess as the answer, in other words, to buzz (Sec 5 L465).
> We compute the Expected Win (\ew{}) scores, following \citet{rodriguez2019quizbowl}.
> \ew{} is the expected probability of winning against an average player as a function of buzzing position and empirically estimated based on the "typical" human gameplay. The \ew{} probablity is only added when the buzzer buzzes for the first time and the answer is correct as in an actual game (Sec 5 L471).
> In our setup,  however, there is only one player, so the score calculation under the competitive setting may be strict. Therefore, we also calculate \ew{} with an \oracle{} buzzer (\ewo{}) where the buzzer would buzz as soon as the guesser gets the correct answer. (Sec 5, footnote9)
> Full input accuracy means the accuracy given the full text of the question.
>
>
> R4) “Both a pro and con the explicitation is tied to structured fields from wikipedia.”
> → Our proposed method of automating explicitation is indeed grounded in structured data. It enables precise control over deciding and generating explicitation by benefiting from consistency and enrichment of the data while avoiding hallucinations as LLMs. However, this approach may suffer from rigidity and limited creativity, thus having less flexibility on dealing with diverse natural language input, which could lower the quality of explicitation.
> We will investigate approaches that integrate structured data and language models in future work.

---

### Meta-Review · Area_Chair_GHVB · 2023-09-20

**Recommendation:** 4

**Metareview:**

The authors focus on explicitation in translation, which explains the missing context by considering cultural differences between source and target audiences.
They first create a dataset of naturally occurring explitations, WIKIEXPL, that they collect from Wikipedia and annotate with human translators.
Inspired by WIKIEXPL, they propose a technique to generate explicitation automatically.
They show the effectiveness of the proposed automatic explicitation method, both intrinsic evaluation by human evaluators and extrinsic evaluation by multilingual question answering tasks.

Many reviewers agree that WIKIEXPL is a valuable resource for research on translation and that there is much to learn from how they create it.
One reviewer is not impressed with the proposed automatic explicitation method because it simply extracts some structured fields from Wikipedia.

Although none of the reviewers pointed this out, the meta-reviewer thinks the proposed method has something in common with the following work, which generates a description for unknown phrases to bridge the knowledge cap.

[Ishiwatari+, NAACL-2019] Learning to Describe Unknown Phrases in Local and Global Contexts

---

### Decision · Program_Chairs · 2023-10-07

**Decision:**

Accept-Main

**Comment:**

The authors focus on explicitation in translation, which explains the missing context by considering cultural differences between source and target audiences.
They first create a dataset of naturally occurring explitations, WIKIEXPL, that they collect from Wikipedia and annotate with human translators.
Inspired by WIKIEXPL, they propose a technique to generate explicitation automatically.
They show the effectiveness of the proposed automatic explicitation method, both intrinsic evaluation by human evaluators and extrinsic evaluation by multilingual question answering tasks.

Many reviewers agree that WIKIEXPL is a valuable resource for research on translation and that there is much to learn from how they create it.
One reviewer is not impressed with the proposed automatic explicitation method because it simply extracts some structured fields from Wikipedia.

Although none of the reviewers pointed this out, the meta-reviewer thinks the proposed method has something in common with the following work, which generates a description for unknown phrases to bridge the knowledge cap.

[Ishiwatari+, NAACL-2019] Learning to Describe Unknown Phrases in Local and Global Contexts